# Hepatitis B vaccination status and knowledge, attitude, and practice regarding Hepatitis B among preclinical medical students of a medical college in Nepal

Dhan Bahadur Shrestha[1]*, Manita Khadka[2], Manoj Khadka[2], Prarthana Subedi[2], Subashchandra Pokharel[2], Bikash Bikram Thapa[2]

1 Mangalbare Hospital, Morang, Nepal, 2 Nepalese Army Institute of Health Sciences (NAIHS), Shree Birendra Hospital, Chhauni, Kathmandu, Nepal

* medhan75@gmail.com

## Abstract

### Background

Hepatitis B imposes a major public health problem with an increased risk of occupational exposure among unvaccinated health care workers. This study was conducted to determine the Hepatitis B vaccination status, along with the knowledge, attitude, and practice regarding Hepatitis B, among preclinical medical students of a medical college in Nepal.

### Materials and methods

This descriptive study was conducted among preclinical students of a medical college in Kathmandu, Nepal from 6th July to 14th July 2020. The whole sampling technique was used. Data were collected using a pretested, self-administered questionnaire which was emailed to individuals and analyzed with the statistical package for social sciences version-22.

### Results

A total of 181 students participated in the study out of 198, giving a response rate of 91.4%. Among the study participants, only 67 (37%) were fully vaccinated against Hepatitis B while 71 (39.2%) were never vaccinated. For the majority (74.6%) of the non-vaccinated participants, the main reason for not getting vaccination was a lack of vaccination programs. Half the study participants (n = 92, 50.8%) had good knowledge, attitude and practice regarding hepatitis B. The median knowledge, attitude and practice scores towards Hepatitis B were 61.00 (57.00–66.00), 20(18.00–21.00) and 21(19.00–23.00) respectively.

### Conclusions

The majority of preclinical medical students were not fully vaccinated against Hepatitis B and only half of them had acceptable knowledge, attitude and practice towards Hepatitis B, which makes them vulnerable to the infection. This might represent the situation of not only Nepal, but also all South Asian countries, and creates concern about whether students take

**Data Availability Statement:** All relevant data are within the manuscript and/or Supporting Information files.

**Funding:** The author(s) received no specific funding for this work.

**Competing interests:** The authors have declared that no competing interests exist.

the vaccination programs seriously. Since unavailability of vaccination program is the main cause of non-vaccination, we strongly recommend the provision of the Hepatitis B vaccination program to the preclinical medical students.

## Introduction

Hepatitis B virus is the most contagious blood-borne pathogen that can cause both acute and chronic liver diseases [1, 2]. Vaccination is the mainstay of prevention against Hepatitis B infection with 90%-100% protection conferred following complete vaccination [2, 3]. World Health Organization (WHO) recommends special consideration of healthcare workers & medical students for Hepatitis B virus screening and vaccination [2].

The risk of accidental exposure among medical students is high due to a lack of experience and direct contact with a patient's potentially infectious body fluids [4–6]. A cross sectional study conducted at two medical schools in Munich showed 23% lifetime prevalence of needle stick injuries among medical students especially during blood taking practices in clinical years [6]. Hence, medical students at the start of clinical years are at high risk of Hepatitis B infection. However, no studies in the past assessed Hepatitis B vaccination status among preclinical level medical students in Nepal.

In Nepal, Hepatitis B vaccine was introduced in 2002 and it is given to infants only at the age of six, ten and fourteen weeks as per National Immunization Schedule of Nepal. The government lacks program to vaccinate high risk population like health care workers and medical students [7]. In line with the vaccination recommendation by WHO to high risk population, health workers should be vaccinated in a timely manner. There is no alternative to vaccines for effective protection.

To provide a basis for the implementation of Hepatitis B vaccination programs, our study aims to estimate the Hepatitis B vaccination status along with knowledge, attitude and practice regarding Hepatitis B infection among preclinical medical students of a medical college in Nepal.

## Materials and methods

### Study design and settings

This is a descriptive cross-sectional study done among the preclinical year (1st and 2nd year) students of the Nepalese Army Institute of Health Sciences (NAIHS), Nepal. The data for the study were collected from 6th July 2020 to 14th July 2020. The whole sampling technique was used. Students who were no longer attending classes and were not in contact were excluded from the study. This made the population size to be 198 (first year-100, second year-98). The data were collected from the participants via Google forms sent out by email after explaining the objective of the study in the form itself. The participation was completely voluntary and the anonymity was insured. The participants didn't receive any incentives.

### Study sample

The total population of the preclinical year of NAIHS was taken. All of the participants were reached in 4 phases by making 8 groups of 25 people each; through online sessions. We informed the participants about our study objectives and its implications during the session. The expected non-response rate was kept at 5% of the total population of the study. During

this survey period, all students were stranded at home due to the Coronavirus disease 2019 (COVID-19) pandemic imposed lockdown. Also Nepal government and global health governing agencies like WHO were advocating to limit gathering and movements due to fear of the spread of COVID-19. So we emailed a questionnaire to participants via Google forms after an online session.

## Study instrument

A self-administered online questionnaire containing 21 items was used for the study (S1 File). It contained 3 items for demographics, 5 for the Knowledge section, 5 for the attitude and 5 for practice, 1 for vaccination status and 1 for total dose taken, and 1 for the reason of not getting vaccinated. The questionnaire was developed after an extensive literature search in the English language. The questionnaire was pre-tested among 5% of the study sample, 10 students from the 3rd and 4th year of medical school, modified accordingly, and then sent to the study participants.

## Participant characteristics

The survey questionnaire included baseline exposure variables like age (below 19 years, 20 years and above), sex, vaccination status, and academic year (1st year, 2nd year).

## Vaccination relevant variables

A semi-structured pre-piloted knowledge, attitude, and practice questionnaire was used to collect data, and its internal consistency was assessed by using Cronbach's α. The internal reliability of the present study was found to be 0.698 indicating high internal consistency for our scale for this specific sample. Five-point Likert scale was used to quantify the response on knowledge, attitude, and practice, with 1 being the least acceptable response (for that particular question based on available scientific understanding) to the item asked and 5 being the most acceptable response to that item. Selected questions in the knowledge section [questions 2(d), 2(f), 2(g), 5(b), and 5(d)] and attitude section (question 6 and 10) were reverse scored while rest were scored 1–5 from strongly disagree to strongly agree. Knowledge related to Hepatitis B was assessed under five questions with some addition subquestions, giving a possible score range from 15–75, while there were five questions for the attitude and practice sections, making possible score ranges from 5–25.

## Statistical methods

The data collected through the Google forms were extracted to Microsoft Excel-13, then imported and analyzed by using SPSS (Statistical Package for Social Sciences) version 22. Kolmogorov-Smirnov (K-S) test and the Shapiro-Wilk test were used to assess the normality of the data distribution and the data distribution was classified as normal if the significant value of the test is greater than 0.05. If the value is below 0.05, the data are classified as non-normal distribution. Our data were non-normal so non-parametric tests were used using the median as a measure of distribution. A Chi-square test was used to check the association between variables. Spearman's rho was used to check the correlation between total scores of knowledge and attitude, knowledge and practice, and attitude and practice.

## Ethical consideration

The study was approved by the Institutional Review Committee of NAIHS. All the participants were informed about the study and its objective by incorporating the consent form in the questionnaire itself. Anyone who filled the form was understood to have given the consent.

## Results

### General characteristics

Table 1 shows the general characteristics of the respondents who participated in the study. Out of 198 preclinical medical students, 181 (91.4% response rate) students participated in the study, 85 (47%) from the first year, and 96 (53%) from the second year. The mean age of the study participants was 19.93 (±1.436) years, ranging from 17–28 years. The majority of the respondents were male (n = 123, 68%). Regarding Hepatitis B vaccination status, 71 (39.2%) study participants weren't vaccinated while 110 (60.8%) students were vaccinated. Among the 110 vaccinated students, 67 (60.9%) received at least 3 doses of Hepatitis B while 43 (39.1%) received less than 3 doses. Among 71 non-vaccinated students, the major reason for not being vaccinated was the lack of a vaccination program being offered (n = 53, 74.6%).

### Assessment of knowledge related to Hepatitis B

The majority of the study participants agreed that Hepatitis B is caused by a virus (87.3% strongly agreed) and Hepatitis B can cause liver cancer (43.6% strongly agreed, 37% agreed). In terms of knowledge on the mode of transmission, around three fourths of the respondents strongly agreed on contaminated blood and body fluids (78.5%), unsterilized syringes/needles (75.7%), unprotected sex (71.8%) and infected mother to fetal transmission (71.8%). Similarly, most of the respondents disagreed on casual contact (49.2% strongly disagreed, 26.5% disagreed) and cough/sneeze (40.9% strongly disagreed, 22.7% disagreed) as a mode of transmission. However, regarding contaminated food/water as a mode of transmission for Hepatitis B, only 33.7% strongly disagreed while 22.1% were neutral and 11.6% strongly agreed. About two thirds agreed that healthcare workers are at increased risk of getting Hepatitis B (29.8% strongly agreed, 38.1% agreed). When asked about prevention, the majority responded with vaccination (80.1% strongly

**Table 1. Baseline characteristics and Hepatitis B vaccination status of the respondents (N = 181).**

| Variables | | N | % |
|---|---|---|---|
| Age (in Years) | 19 and below | 67 | 37.0 |
| | 20 and above | 114 | 63.0 |
| | *Mean±SD* | 19.93±1.436 | |
| Gender | Male | 123 | 68.0 |
| | Female | 58 | 32.0 |
| Academic year | 1st Year MBBS | 85 | 47.0 |
| | 2nd Year MBBS | 96 | 53.0 |
| Vaccinated against hepatitis B | No | 71 | 39.2 |
| | Yes | 110 | 60.8 |
| Doses of Hepatitis B vaccine received | Not vaccinated | 71 | 39.2 |
| | One | 14 | 7.7 |
| | Two | 29 | 16.0 |
| | Three | 55 | 30.4 |
| | More than three | 12 | 6.6 |
| Reason for not being vaccinated against Hepatitis B | No vaccination program offered | 53 | 29.3 |
| | Low risk of Hepatitis B | 6 | 3.3 |
| | Not sure about vaccination status | 4 | 2.2 |
| | Lack of Knowledge | 4 | 2.2 |
| | High vaccination fees | 2 | 1.1 |
| | Efficacy doubted | 2 | 1.1 |

**Table 2. Knowledge, attitude and practice findings of the respondents.**

| Questions for response | Strongly Disagree n (%) | Disagree n (%) | Neutral n (%) | Agree n (%) | Strongly agree n (%) |
|---|---|---|---|---|---|
| **Assessment of Knowledge related to Hepatitis B** | | | | | |
| 1. Hepatitis B is caused by a virus. | 10 (5.5) | 4 (2.2) | 0 (0) | 9 (5.0) | 158 (87.3) |
| 2. Hepatitis B can be transmitted by: a. Infected mother to fetus | 2 (1.1) | 3 (1.7) | 9 (5.0) | 37 (20.4) | 130 (71.8) |
| 2. b. Contaminated blood and body fluids | 2 (1.1) | 1 (0.6) | 2 (1.1) | 34 (18.8) | 142 (78.5) |
| 2. c. Unprotected sex | 7 (3.9) | 5 (2.8) | 10 (5.5) | 29 (16.0) | 130 (71.8) |
| 2. d. Casual contact (shaking hands) | 89 (49.2) | 48 (26.5) | 18 (9.9) | 16 (8.8) | 10 (5.5) |
| 2. e. Unsterilized syringes/needles | 3 (1.7) | 2 (1.1) | 4 (2.2) | 35 (19.3) | 137 (75.7) |
| 2. f. Coughing/sneezing | 74 (40.9) | 41 (22.7) | 33 (18.2) | 19 (10.5) | 14 (7.7) |
| 2. g. Contaminated food/water | 61 (33.7) | 30 (16.6) | 40 (22.1) | 29 (16.0) | 21 (11.6) |
| 3. Hepatitis B can cause liver cancer. | 4 (2.2) | 8 (4.4) | 23 (12.7) | 67 (37.0) | 79 (43.6) |
| 4. Healthcare workers are at increased risk of getting hepatitis B than general population: | 5 (2.8) | 16 (8.8) | 37 (20.4) | 69 (38.1) | 54 (29.8) |
| 5. Hepatitis B can be prevented by: a. Vaccination | 2 (1.1) | 2 (1.1) | 3 (1.7) | 29 (16.0) | 145 (80.1) |
| 5. b. Antivirals | 6 (3.3) | 7 (3.9) | 45 (24.9) | 75 (41.4) | 48 (26.5) |
| 5. c. Avoiding sharp needle/syringe injury | 5 (2.8) | 10 (5.5) | 18 (9.9) | 76 (42.0) | 72 (39.8) |
| 5. d. Avoiding contaminated food/water | 37 (20.4) | 34 (18.8) | 46 (25.4) | 43 (23.8) | 21 (11.6) |
| 5. e. Using gloves when handling body fluids | 1 (0.6) | 5 (2.8) | 14 (7.7) | 76 (42.0) | 85 (47.0) |
| **Assessment of Attitude towards Hepatitis B** | | | | | |
| 6. I feel uncomfortable sitting with a hepatitis B infected person. | 34 (18.8) | 45 (24.9) | 55 (30.4) | 35 (19.3) | 12 (6.6) |
| 7. I don't mind shaking hands/hugging with a hepatitis B infected person. | 12 (6.6) | 26 (14.4) | 41 (22.7) | 65 (35.9) | 37 (20.4) |
| 8. I believe the hepatitis B vaccine is safe and effective. | 5 (2.8) | 2 (1.1) | 17 (9.4) | 92 (50.8) | 65 (35.9) |
| 9. I believe healthcare workers should receive hepatitis B vaccination. | 0 (0) | 0 (0) | 13 (7.2) | 49 (27.1) | 119 (65.7) |
| 10. I don't need hepatitis B vaccination because I am not at risk | 101 (55.8) | 54 (29.8) | 19 (10.5) | 6 (3.3) | 1 (0.6) |
| **Assessment of Practice towards Hepatitis B** | | | | | |
| 11. I ask/use a new blade for shaving/hair cutting. | 4 (2.2) | 4 (2.2) | 11 (6.1) | 33 (18.2) | 129 (71.3) |
| 12. I ask for a new syringe before injection. | 0 (0) | 1 (0.6) | 6 (3.3) | 22 (12.2) | 152 (84.0) |
| 13. I ask for sterilized equipment for ear/nose piercing. | 3 (1.7) | 2 (1.1) | 18 (9.9) | 48 (26.5) | 110 (60.8) |
| 14. I always report for needle prick / sharp injuries. | 0 (0) | 18 (9.9) | 46 (25.4) | 59 (32.6) | 58 (32.0) |
| 15. I attend hepatitis B related awareness programs. | 15 (8.3) | 21 (11.6) | 65 (35.9) | 50 (27.6) | 30 (16.6) |

agreed), avoiding sharp needles/syringes (39.8% strongly agreed, 42% agreed), and using gloves when handling body fluids (47% strongly agreed, 42% agreed). However, regarding antivirals as a preventive measure, only 3.3% strongly disagreed while 24.9% were neutral and 26.5% strongly agreed. Additionally, 20.4% strongly disagreed on avoiding contaminated food/water as a preventive measure for Hepatitis B while 25.4% were neutral and 11.6% strongly agreed (Table 2). The median score for knowledge was 61 (Table 3).

## Assessment of attitude towards Hepatitis B

The majority were neutral (n = 55, 30.4%) toward sitting with a Hepatitis B positive person, however, 6.6% (n = 12) strongly agreed on feeling uncomfortable while sitting with a Hepatitis

**Table 3. Summation of knowledge, attitude and practice score distribution.**

| | Knowledge sum (n = 181) | Attitude sum (n = 181) | Practice Sum (n = 181) | Total Score (n = 181) |
|---|---|---|---|---|
| Mean | 60.86 | 19.91 | 20.97 | 101.73 |
| Median | 61.00 | 20.00 | 21.00 | 102.00 |
| IQR | (57.00–66.00) | (18.00–21.00) | (19.00–23.00) | (96.00–108.00) |

B positive person and 18.8% (n = 34) strongly disagreed. Many don't mind shaking hands with a Hepatitis B positive person (n = 65, 35.9%), however, 6.6% (n = 12) strongly disagreed on doing so. Around half of the respondents, 50.8% (n = 92) agreed that hepatitis B vaccination is safe and effective whereas 2.8% (n = 5) strongly disagreed. Most of the respondents, 65.7% (n = 119) strongly agreed that healthcare workers should receive hepatitis B vaccination and 55.8% (n = 101) strongly agreed that they need Hepatitis B vaccination because they are at risk (Table 2). The median attitude score was 20 (Table 3).

## Assessment of practice towards Hepatitis B

Among the respondents, the majority (n = 129, 71.3%) ask for a new blade while cutting or shaving hair. 84% (n = 152) of them ask for a new syringe before injection. 32% (n = 58) strongly agreed on reporting their needle prick injury whereas 9.9% (n = 18) disagreed. A small number of people (n = 15, 8.3%) strongly disagreed on attending any hepatitis B awareness program whereas 27.6% (n = 50) agreed (Table 2). The median score for practice was 21 (Table 3).

## Categorization of knowledge attitude and practice (KAP) score and its association with baseline characteristics

Table 4 demonstrates the association of KAP score with baseline characteristics like age, gender, academic year, vaccination status, doses of vaccination received, and reasons for not being vaccinated. There is no significant association observed. The median KAP score was 102 (Table 3). Fig 1 shows that the KAP was good (above the median score) among 92 (50.8%) respondents.

**Table 4. Association of KAP score with baseline characteristics.**

| Variables | | KAP score Category | | p-value |
|---|---|---|---|---|
| | | In-adequate (<102) | Good (≥102) | |
| Gender: | Male | 62 | 61 | .628 |
| | Female | 27 | 31 | |
| Age | 19 and below | 36 | 31 | .347 |
| | 20 and above | 53 | 61 | |
| Academic year | 1st Year MBBS | 44 | 41 | .511 |
| | 2nd Year MBBS | 45 | 51 | |
| Vaccinated against hepatitis B | No | 37 | 34 | .525 |
| | Yes | 52 | 58 | |
| Doses of Hepatitis B vaccine | No | 37 | 34 | .455 |
| | One | 8 | 6 | |
| | Two | 10 | 19 | |
| | Three | 29 | 26 | |
| | More than three | 5 | 7 | |
| Reason for Not vaccinated against Hepatitis B | No vaccination program offered | 27 | 26 | .564 |
| | Low risk of Hepatitis B | 3 | 3 | |
| | Not sure about vaccination status | 1 | 3 | |
| | High vaccination fees | 2 | 0 | |
| | Efficacy doubted | 1 | 1 | |
| | Lack of Knowledge | 3 | 1 | |

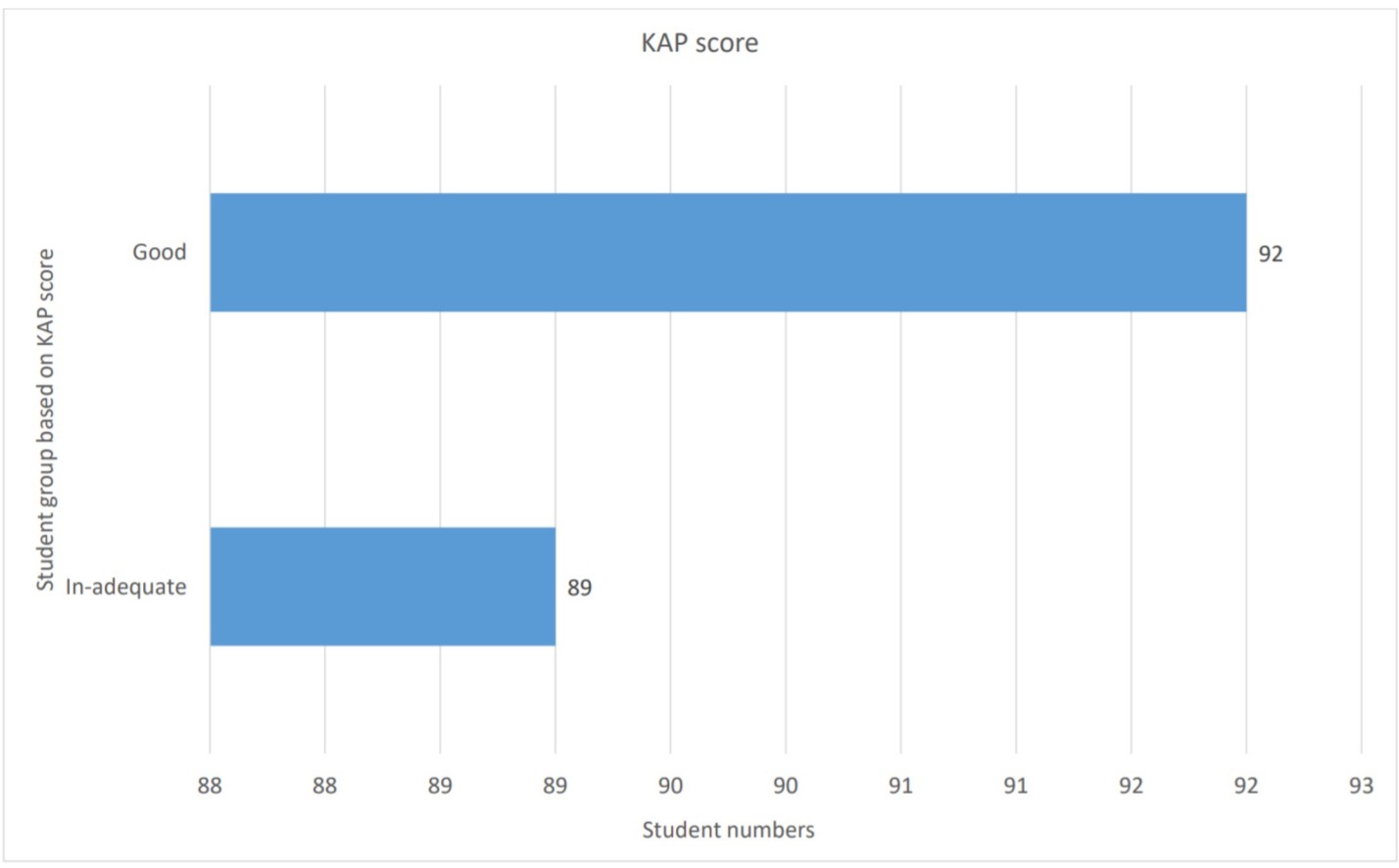

**Fig 1. KAP score category among the respondents.**

## Correlation of knowledge, attitude, and practice on Hepatitis B

We found a weak positive correlation between knowledge with attitude (r = 0.343) and attitude with practice (r = 0.170). However, no significant correlation was found between knowledge with practice (r = 0.009, p-value 0.909) (Table 5).

## Discussion

The risk of exposure to Hepatitis B infection among medical students is the same as, if not greater than, other healthcare workers since they are banked on to be involved in patient care at the beginning of clinical training. This highlights the importance of getting vaccinated and acquiring adequate knowledge about the infection before getting into clinical training.

**Table 5. Correlation of knowledge, attitude, and practice.**

| Variables | Spearman's correlation coefficient | p-value |
|---|---|---|
| Knowledge-Attitude | 0.343* | p< 0.01 |
| Knowledge-Practice | 0.009 | P = 0.909 |
| Attitude-Practice | 0.170** | P = 0.022 |

*Correlation is significant at p< 0.01 level (2-tailed).

**Correlation is significant at the 0.05 level (2-tailed).

However, Nepal lacks adequate studies that assess the vaccination status of the students and their knowledge, attitude, and practice regarding Hepatitis B infection. This study looks into the vaccination status of preclinical students and describes their KAP towards the infection. These students will soon get exposed to clinical settings after the COVID-19 pandemic will come under control.

The Hepatitis B vaccination status among preclinical medical students in our study was 60.8% which is quite lower than the only other study done among medical students in the country where 86.5% of the students were vaccinated [7]. But our finding is similar to a study from Pakistan, where 60% were vaccinated [8]. And it is higher than the findings of a study from Nigeria and another study from Pakistan where 47.7% and 42.2% were reported to have been vaccinated respectively [9, 10]. Among total students, 37.01% (n = 67) were fully vaccinated (3 and more doses). This is similar to the study from Saudi Arabia where 38% were found to have received all three doses, and much higher than 2% completing all three doses according to a finding from Ethiopia [11, 12]. However, the finding from another study done in Nepal showed a higher percentage (83.7%) of students completing full doses [7]. This warrants a need to look into the vaccination status of students before going into clinical years to ensure high vaccination rates during their clinical training.

Among the non-vaccinated participants (39.2%) of our study, the main reasons for non-vaccination were found to be: no vaccination program offered (74.6%) followed by belief there is a low risk of Hepatitis B (8.5%) and lack of knowledge (5.6%). The result is similar to other studies from Nepal and Nigeria that showed a lack of vaccination programs (43.2%) and lack of opportunity (57.4%) respectively as the major reasons for non-vaccination [7, 9]. These findings sufficiently shed light on the urgent necessity to implement vaccination programs for medical students. However, the high cost of vaccines was stated as the major reason for non-vaccination among health science students from Uganda [13].

Out of the students surveyed, 92.3% were aware that Hepatitis B is caused by a virus. A study from Nepal done among preclinical medical and dental students revealed that 93.6% of the participants were aware of the cause of Hepatitis B infection [14]. Regarding the knowledge about the mode of transmission, the majority knew about infected mother to fetus transmission (92.2%), transmission through contaminated blood and body fluids (97.3%), unprotected sex (87.8%), and unsterilized syringes/needles (95%). The findings are parallel to the findings of a study from Northeast Ethiopia and higher than the study from Haramaya University, Ethiopia [12, 15]. However, very few participants strongly disagreed with the transmission of infection via casual contact, coughing/sneezing, and contaminated food/water. This gap in knowledge could increase the probability of patients with hepatitis B infection from being ostracized from society.

80.6% of respondents agreed Hepatitis B virus can cause liver cancer in the present study. This is comparable with Saudi Arabian and Ethiopian studies among medical students showing 75.5% and 81.3% agreeing hepatitis B causes liver cancer respectively [11, 12]. 96.1% of our study participants agreed that the Hepatitis B infection can be prevented by vaccination. The studies from Saudi Arabia and Ethiopia showed 86.5% and 84.6%, respectively, responding that vaccine prevents Hepatitis B [11, 12].

Regarding attitude related to Hepatitis B infection and vaccination; 43.7% didn't feel uncomfortable sitting with a Hepatitis B positive person. 56.3% didn't mind shaking hands or hugging with a Hepatitis B infected person, which is higher than the finding from Saudi Arabia, indicating the more acceptable attitude of students responding to the present study [16]. In our study, 86.7% of the participants thought that the hepatitis B vaccine was safe and effective which is similar to findings from Saudi Arabia and Ethiopia [11, 12]. This finding is higher as compared to another study from Saudi Arabia where only 63% considered the vaccine safe

[16]. 92.8% of our participants believed that healthcare workers should be vaccinated, which is notably higher than in a study from Saudi Arabia [16]. The same study from Saudi Arabia also stated that 62% of its participants believed that they were at higher risk of contracting Hepatitis B infection; which is lower than our study's finding which 85.6% [16]. As of our study, 3.9% thought that they don't need to be vaccinated and that they are not at risk which is comparable with an Indian study's finding with 3.7% responding they don't need it [17]. Though this is a small proportion as compared to other findings it needs to be addressed because medical students, being a part of the healthcare delivery system, should be well aware that they need to be vaccinated as they are always at risk of contracting and spreading the Hepatitis B infection.

In our study, 89.5%, 96.2%, and 87.3% of the participants reported that they asked for a new blade for shaving/hair cutting, a new syringe for injection, and sterile equipment for ear/nose piercing which reflects the good safety practices among the participants. A study showed similar findings in the case of participants asking for a new blade for shaving/hair cutting and sterile equipment for nose/ear piercing [17]. The proportion of respondents who report asking for new syringes for injection in our study is more than that in Ethiopia and India [15, 17]. 64.6% of our study participants said that they always report needle prick/sharp injuries, which is similar to a Saudi Arabian study showing 68% while a study from Ethiopia reported only 53.7% will report needle stick injury [11, 12]. 44.2% of our study participants agreed that they attend Hepatitis B related awareness programs, which is significantly greater than the finding from Ethiopia i.e. 23.9% [15].

This study found that females had better overall KAP scores compared to males, which are per the previous study from Pakistan [18]. However, it contradicts the study from Malaysia which had found no association between gender and knowledge [19]. Similarly, second-year students had better KAP scores than first-year students, which is in line with the findings of a study from Pakistan where 1st professional year had the least knowledge [18]. This can be associated with the fact that second-year students have got more instructions and information and are more aware of the disease. Those who had been vaccinated were found to have better KAP scores than those who weren't. Weak positive correlations of knowledge with attitude and of attitude with practice were found, somewhat similar to the correlations found in a study from Malaysia [19].

The major limitation of this study is that we didn't measure the anti-Hepatitis B surface antibody (HBsAb) titer of the participants; hence the vaccination status could not be verified. Likewise, recall bias, information bias, and social desirability bias might have also affected the result of our study. We had mentioned the date of the introduction of the Hepatitis B vaccine in the immunization program of Nepal in the questionnaire and also collected the responses anonymously to mitigate these biases. This study was done by collecting responses using an online questionnaire in a country where internet penetration is 57% [20]. This may have affected the response rate of our study. Finally, the results may not be generalizable to all the medical colleges of Nepal since the data were collected only from NAIHS but it will surely provide a reference for further research in this field. The health care system needs to be strengthened in poor resource countries like Nepal where a limited number of health care providers are available. Due to the limited availability of resources, health protection and promotion by providers has to be addressed explicitly. Due to the COVID-19 pandemic, the health care delivery system is highly impacted including clinical learning of medical students. Before they get exposed to the clinical rotation, we recommend the governing body to make available and implement mandatory Hepatitis B vaccination to safeguard the health of medical students.

## Conclusion

We found that there were more than one fourths of the respondents not vaccinated for Hepatitis B despite it being a vaccine-preventable disease. The main reason for not getting vaccinated

was the unavailability of vaccination programs. Vaccination programs need to be provided for all healthcare workers and medical students before they go into their clinical years so that they can protect themselves from the hazards of non-vaccination and needle stick injuries. Though Nepal is devoted to the Sustainable Development Goals, there is no national plan to combat Hepatitis B. So, a change must be instilled from the regulatory bodies of medical schools as well as the government and international agencies like WHO and a free vaccination program can be administered to all students coming into the hospital before they enter clinical years.

## Declarations

**Consent for Publication**: All authors granted permission to publish this manuscript.

## Supporting information

**S1 File. Questionnaire.**
(DOCX)

**S1 Anonymous.**
(SAV)

## Author Contributions

**Conceptualization:** Dhan Bahadur Shrestha, Manita Khadka, Manoj Khadka, Prarthana Subedi, Subashchandra Pokharel, Bikash Bikram Thapa.

**Data curation:** Dhan Bahadur Shrestha, Manita Khadka, Manoj Khadka, Prarthana Subedi, Subashchandra Pokharel, Bikash Bikram Thapa.

**Formal analysis:** Dhan Bahadur Shrestha.

**Methodology:** Dhan Bahadur Shrestha, Manita Khadka, Manoj Khadka, Prarthana Subedi, Subashchandra Pokharel, Bikash Bikram Thapa.

**Project administration:** Manita Khadka, Manoj Khadka, Prarthana Subedi, Subashchandra Pokharel.

**Supervision:** Dhan Bahadur Shrestha, Bikash Bikram Thapa.

**Validation:** Dhan Bahadur Shrestha, Manita Khadka, Manoj Khadka.

**Writing – original draft:** Dhan Bahadur Shrestha, Manita Khadka, Manoj Khadka, Prarthana Subedi, Subashchandra Pokharel.

**Writing – review & editing:** Dhan Bahadur Shrestha, Manita Khadka, Manoj Khadka, Prarthana Subedi, Subashchandra Pokharel, Bikash Bikram Thapa.

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
