## [Decision Letter · Decision Letter 0]

6 Oct 2020

PONE-D-20-25582

Hepatitis B vaccination status and knowledge, attitude, and practice regarding Hepatitis B among preclinical medical students of a medical college in Nepal

PLOS ONE

Dear Dr. Shrestha,

Thank you for submitting your manuscript to PLOS ONE. After careful consideration, we feel that it has merit but does not fully meet PLOS ONE’s publication criteria as it currently stands. Therefore, we invite you to submit a revised version of the manuscript that addresses the points raised during the review process.

We look forward to receiving your revised manuscript.

Kind regards,

Isabelle Chemin, PhD

Academic Editor

PLOS ONE

Journal Requirements:

2) Your ethics statement should only appear in the Methods section of your manuscript. If your ethics statement is written in any section besides the Methods, please delete it from any other section.

Reviewers' comments:

Reviewer's Responses to Questions

**Comments to the Author**

1. Is the manuscript technically sound, and do the data support the conclusions?

Reviewer #1: Yes

Reviewer #2: Yes

2. Has the statistical analysis been performed appropriately and rigorously? 

Reviewer #1: Yes

Reviewer #2: Yes

3. Have the authors made all data underlying the findings in their manuscript fully available?

Reviewer #1: Yes

Reviewer #2: No

4. Is the manuscript presented in an intelligible fashion and written in standard English?

Reviewer #1: Yes

Reviewer #2: No

5. Review Comments to the Author

Reviewer #1: This paper reports on a study examining Hepatitis B vaccination status and knowledge, attitudes and practices of Hepatitis B vaccination amongst medical students in Nepal. The manuscript would be strengthened by attention to the following:

1) In the introduction, provide more of an explanation about why it would be important to study medical students’ and their status, knowledge, etc. in Nepal. Does Nepal have a particularly high Hep B infection rate? Would students need to be informed to help provide education to their patients and communities? How would learning about this in the Nepalese context help others who do not live/work/educate students in Nepal. When was the Hep B vaccine introduced in Nepal?

2) I think in the methods the authors intend to state the survey was distributed through e-mail, since it was a Google form that was used.

3) On page 5, there is reference to online sessions regarding study details – can you provide more information about what those were? Were they optional for students/study participants? What information was presented during the session, i.e., purpose of the study and an opportunity to ask questions?

4) While the survey is in a supplemental file (which I could not reach), it would be helpful to provide some more detail about the nature of the questions asked and associated response formats. For the knowledge section, what were the topics, for the attitude section, what were the attitudes/topics, and similarly for the practice questions. What were the options for not getting vaccinated. What were the demographic questions.

5) With respect to the discussion of the findings, would not the medical school curriculum present information about how Hep B is transmitted to students? The respondents in general appear knowledgeable about the cause and mode of transmission, however, given their education as future physicians, I would hope they would all get those questions “correct.” When is infectious disease taught in the curriculum and is Hep B a particular topic?

6) In the discussion and conclusion, address why a reader outside of Nepal would be interested/need to know this information. This relates back to the first suggestion. The authors do include information about studies from other countries in the discussion which is a strength of the discussion. The authors also advocate that vaccination programs should be available for health care workers. While such advocacy is clearly at the national level, would it be helpful if global organizations, such as the World Health Organization, advocated this and/or perhaps provided resources for these programs? I don’t know – I am just speculating and trying to help the authors have their work make more of an impact than at the Nepalese national level.

Reviewer #2: I noted a great deal of simply awkward word choices, and corrected most of them in a copy of the pdf that I added "comments" to. This is not how I usually do it. (I usually copy the sentence and propose the changes in a separate word file, which I then paste the contents of in this box.)

But since I was using that format, I also made 3 or 4 remarks that require some additional writing. I will sommarize them here.

1. Acknowledge that your respondents probably wanted to look good in your eyes, and their teachers' eyes, and their own eyes. "social desirability". So instead of saying "x% of them report needle sticks", say "x% of them SAY THAT they report needle sticks." for example.

I don't know what are the right answers to the knowledge test, what are appropriate attitudes for a physician, and what are the recommended behaviors. You do not identify in the text which items are true and which false, and you do not discuss how you treat the numerical scores on those items. I hope you did not add them all up, true and false together!

You present some summary data - sums of the ratings on items about attitudes, about knowledge. But you do not say what the maximum possible score would be. Are these numbers comparable? The reader can not judge if students are good at knowledge, but poor at translating the knowledge into habitual practice (for example), if you don't say the ranges of the scales. Also, you report an overall scale that combines knowledge, attitude, and behavior into one numerical scale. How should we interpret that? Are they equally weighted? on what basis?

6. PLOS authors have the option to publish the peer review history of their article (what does this mean?). If published, this will include your full peer review and any attached files.

Reviewer #1: No

Reviewer #2: **Yes: **Robert M. Hamm

---

## [Author Response · Author response to Decision Letter 0]

14 Oct 2020

Responses to Reviewer's comments and Questions

Reply: PLOS ONE style followed

2) Your ethics statement should only appear in the Methods section of your manuscript. If your ethics statement is written in any section besides the Methods, please delete it from any other section.

Reply: Ethics statement removed from other area apart from Methods section

Reviewers' comments:

Comments to the Author

1. Is the manuscript technically sound, and do the data support the conclusions?

Reviewer #1: Yes

Reviewer #2: Yes

Reply: NA

2. Has the statistical analysis been performed appropriately and rigorously?

Reviewer #1: Yes

Reviewer #2: Yes

Reply: NA

3. Have the authors made all data underlying the findings in their manuscript fully available?

Reviewer #1: Yes

Reviewer #2: No

Reply: All data collected were analyzed and made available in the manuscript. No part of data left un-analyzed 

4. Is the manuscript presented in an intelligible fashion and written in Standard English?

Reviewer #1: Yes

Reviewer #2: No

Reply: Any language, typographical, presentation or grammatical errors commented and raised by reviewers revisited and rectified. Also all authors review papers to pick any such errors to avoid it.

5. Review Comments to the Author

Reviewer #1: This paper reports on a study examining Hepatitis B vaccination status and knowledge, attitudes and practices of Hepatitis B vaccination amongst medical students in Nepal. The manuscript would be strengthened by attention to the following:

1) In the introduction, provide more of an explanation about why it would be important to study medical students’ and their status, knowledge, etc. in Nepal. Does Nepal have a particularly high Hep B infection rate? Would students need to be informed to help provide education to their patients and communities? How would learning about this in the Nepalese context help others who do not live/work/educate students in Nepal. When was the Hep B vaccine introduced in Nepal?

Reply: These issues rectified by adding and modifying content in introduction section to give lucid idea of Hepatitis B in context of Nepal and its utility in other part of world. 

2) I think in the methods the authors intend to state the survey was distributed through e-mail, since it was a Google form that was used.

Reply: All participants were informed and detailed about the aims of study via online group discussion dividing all two batches into smaller groups, after that all individuals were requested to fill form made available as Google form and link shared to individuals through email. 

3) On page 5, there is reference to online sessions regarding study details – can you provide more information about what those were? Were they optional for students/study participants? What information was presented during the session, i.e., purpose of the study and an opportunity to ask questions?

Reply: Those session were conducted to provide detail information regarding study objectives and implications to the participants and to answer any questions related to study.

4) While the survey is in a supplemental file (which I could not reach), it would be helpful to provide some more detail about the nature of the questions asked and associated response formats. For the knowledge section, what were the topics, for the attitude section, what were the attitudes/topics, and similarly for the practice questions? What were the options for not getting vaccinated? What were the demographic questions?

Reply: Questions were made available as supplemental file and those questions were analyzed and kept in tables of result sections respectively. 

5) With respect to the discussion of the findings, would not the medical school curriculum present information about how Hep B is transmitted to students? The respondents in general appear knowledgeable about the cause and mode of transmission, however, given their education as future physicians, I would hope they would all get those questions “correct.” When infectious disease is taught in the curriculum and is Hep B a particular topic?

Reply: Based on curriculum of Microbiology subject in Tribhuvan university (where NAIHS is affiliated), during first and second year of preclinical days infectious disease is taught to medical students. 

6) In the discussion and conclusion, address why a reader outside of Nepal would be interested/need to know this information. This relates back to the first suggestion. The authors do include information about studies from other countries in the discussion which is a strength of the discussion. The authors also advocate that vaccination programs should be available for health care workers. While such advocacy is clearly at the national level, would it be helpful if global organizations, such as the World Health Organization, advocated this and/or perhaps provided resources for these programs? I don’t know – I am just speculating and trying to help the authors have their work make more of an impact than at the Nepalese national level.

Reply: Relevant discussion in introduction, discussion and conclusion section added/modified to address above issues. Also at present tourism and global migration is increased so it is advisable to get vaccinated for Hep B among high risk groups. 

Reviewer #2: I noted a great deal of simply awkward word choices, and corrected most of them in a copy of the pdf that I added "comments" to. This is not how I usually do it. (I usually copy the sentence and propose the changes in a separate word file, which I then paste the contents of in this box.)

But since I was using that format, I also made 3 or 4 remarks that require some additional writing. I will sommarize them here.

1. Acknowledge that your respondents probably wanted to look good in your eyes, and their teachers' eyes, and their own eyes. "social desirability". So instead of saying "x% of them report needle sticks", say "x% of them SAY THAT they report needle sticks." for example.

Reply: Implemented and rectified as appropriate. 

I don't know what the right answers to the knowledge test are, what appropriate attitudes are for a physician, and what the recommended behaviors are. You do not identify in the text which items are true and which false, and you do not discuss how you treat the numerical scores on those items. I hope you did not add them all up, true and false together!

Reply: These issues discussed in more detail and in lucid ways in method section. 

You present some summary data - sums of the ratings on items about attitudes, about knowledge. But you do not say what the maximum possible score would be. Are these numbers comparable? The reader cannot judge if students are good at knowledge, but poor at translating the knowledge into habitual practice (for example), if you don't say the ranges of the scales. Also, you report an overall scale that combines knowledge, attitude, and behavior into one numerical scale. How should we interpret that? Are they equally weighted? on what basis?

Reply: We authors found that score scale were missing so added in method section while revising the draft.

---

## [Decision Letter · Decision Letter 1]

3 Nov 2020

PONE-D-20-25582R1

Hepatitis B vaccination status and knowledge, attitude, and practice regarding Hepatitis B among preclinical medical students of a medical college in Nepal

PLOS ONE

Dear Dr. Shreshta,

Thank you for submitting your manuscript to PLOS ONE. After careful consideration, we feel that it has merit but does not fully meet PLOS ONE’s publication criteria as it currently stands. Therefore, we invite you to submit a revised version of the manuscript that addresses the minor points raised during the review process.

We look forward to receiving your revised manuscript.

Kind regards,

Isabelle Chemin, PhD

Academic Editor

PLOS ONE

Reviewers' comments:

Reviewer's Responses to Questions

**Comments to the Author**

1. If the authors have adequately addressed your comments raised in a previous round of review and you feel that this manuscript is now acceptable for publication, you may indicate that here to bypass the “Comments to the Author” section, enter your conflict of interest statement in the “Confidential to Editor” section, and submit your "Accept" recommendation.

Reviewer #2: (No Response)

2. Is the manuscript technically sound, and do the data support the conclusions?

Reviewer #2: Yes

3. Has the statistical analysis been performed appropriately and rigorously? 

Reviewer #2: Yes

4. Have the authors made all data underlying the findings in their manuscript fully available?

Reviewer #2: No

5. Is the manuscript presented in an intelligible fashion and written in standard English?

Reviewer #2: Yes

6. Review Comments to the Author

Reviewer #2: My suggestions are basically improvements in English usage. I use my usual method, cut and paste, rather than tags in a pdf.

I think you should reconsider your answer to “data made available”. I think that means that your raw data (suitably anonymized) could be provided to another researcher who might want to check your analyses. You have asserted it is available because you showed the results in your tables, which misses the point. I defer to the editors to explain what they mean by “data made available”.

The figure at the end is so simple, it seems unnecessary. Yes, there are two numbers, and they are different from each other. That could be adequately conveyed with a sentence in the text. But if the editor is happy with it, let it be.

In “medical schools in Munich showed 23 % lifetime prevalence” and at least one other place, remove the space between the number and the % sign.

“was introduced in 2002 AD”. We know it is AD, common era (CE). Don’t need to say it.

“health workers

59 should be vaccinated timely.” Should be “health workers should be vaccinated in a timely manner” in order to be fully grammatically correct.

“There is no replacement to vaccines for effective protection.” Perhaps “no alternative” rather than “replacement”.

P 6. Change “Selected questions in knowledge section” to “Selected questions in the knowledge section”

Change “Knowledge related to Hepatitis B was assessed under five questions with some specific questions giving possible score ranged from 15-75” to “Knowledge related to Hepatitis B was assessed under five questions with some addition subquestions, giving a possible score range from 15-75”. I think that is what you mean. Maybe there were five questions but with 1a, 1b, … 5b, 5c or something.

Change “while questions for attitude and practice section were five making possible score range from 5-25” to “while there were five questions for the attitude and practice sections, making possible score ranges from 5-25”.

P 7. Change “to check the correlation between total score among knowledge and attitude, knowledge and practice, and attitude and practice” to “to check the correlation between total scores of knowledge and attitude, knowledge and practice, and attitude and practice”.

P 16. Change “The studies from Saudi Arabia and Ethiopia showed 86.5% and 84.6% respectively responded that vaccine prevents Hepatitis B” to “The studies from Saudi Arabia and Ethiopia showed 86.5% and 84.6%, respectively, responding that vaccine prevents Hepatitis B”.

P 18. Change “recall bias, information bias and, social desirability bias might have also affected” to ““recall bias, information bias, and social desirability bias might have also affected”. (move the comma)

P 19. I think “consent for publication” means that all the authors agree that it can be published, with their name on it; but maybe not.

Acknowledgements – was there no one who helped with the study, typing things up, or entering data, or facilitating access? Minor things not meriting authorship.

7. PLOS authors have the option to publish the peer review history of their article (what does this mean?). If published, this will include your full peer review and any attached files.

Reviewer #2: **Yes: **Robert M. Hamm

---

## [Author Response · Author response to Decision Letter 1]

7 Nov 2020

Response to reviewer

Comments to the Author

1. If the authors have adequately addressed your comments raised in a previous round of review and you feel that this manuscript is now acceptable for publication, you may indicate that here to bypass the “Comments to the Author” section, enter your conflict of interest statement in the “Confidential to Editor” section, and submit your "Accept" recommendation.

Reviewer #2: (No Response)

Authors Reply: Not applicable 

2. Is the manuscript technically sound, and do the data support the conclusions?

Reviewer #2: Yes

Authors Reply: Not applicable 

3. Has the statistical analysis been performed appropriately and rigorously?

Reviewer #2: Yes

Authors Reply: Not applicable 

4. Have the authors made all data underlying the findings in their manuscript fully available?

Reviewer #2: No

Authors Reply: Made available 

5. Is the manuscript presented in an intelligible fashion and written in standard English?

Reviewer #2: Yes

Authors Reply: Not applicable 

6. Review Comments to the Author

Reviewer #2: My suggestions are basically improvements in English usage. I use my usual method, cut and paste, rather than tags in a pdf.

I think you should reconsider your answer to “data made available”. I think that means that your raw data (suitably anonymized) could be provided to another researcher who might want to check your analyses. You have asserted it is available because you showed the results in your tables, which misses the point. I defer to the editors to explain what they mean by “data made available”.

Authors Reply: anonymized data submitted 

The figure at the end is so simple, it seems unnecessary. Yes, there are two numbers, and they are different from each other. That could be adequately conveyed with a sentence in the text. But if the editor is happy with it, let it be.

In “medical schools in Munich showed 23 % lifetime prevalence” and at least one other place, remove the space between the number and the % sign.

Authors Reply: Rectified

“was introduced in 2002 AD”. We know it is AD, common era (CE). Don’t need to say it.

“health workers

Authors Reply: Rectified

59 should be vaccinated timely.” Should be “health workers should be vaccinated in a timely manner” in order to be fully grammatically correct.

“There is no replacement to vaccines for effective protection.” Perhaps “no alternative” rather than “replacement”.

Authors Reply: Rectified

P 6. Change “Selected questions in knowledge section” to “Selected questions in the knowledge section”

Authors Reply: Rectified

Change “Knowledge related to Hepatitis B was assessed under five questions with some specific questions giving possible score ranged from 15-75” to “Knowledge related to Hepatitis B was assessed under five questions with some addition subquestions, giving a possible score range from 15-75”. I think that is what you mean. Maybe there were five questions but with 1a, 1b, … 5b, 5c or something.

Authors Reply: Rectified

Change “while questions for attitude and practice section were five making possible score range from 5-25” to “while there were five questions for the attitude and practice sections, making possible score ranges from 5-25”.

Authors Reply: Rectified

P 7. Change “to check the correlation between total score among knowledge and attitude, knowledge and practice, and attitude and practice” to “to check the correlation between total scores of knowledge and attitude, knowledge and practice, and attitude and practice”.

Authors Reply: Rectified

P 16. Change “The studies from Saudi Arabia and Ethiopia showed 86.5% and 84.6% respectively responded that vaccine prevents Hepatitis B” to “The studies from Saudi Arabia and Ethiopia showed 86.5% and 84.6%, respectively, responding that vaccine prevents Hepatitis B”.

Authors Reply: Rectified

P 18. Change “recall bias, information bias and, social desirability bias might have also affected” to ““recall bias, information bias, and social desirability bias might have also affected”. (move the comma)

Authors Reply: Rectified

P 19. I think “consent for publication” means that all the authors agree that it can be published, with their name on it; but maybe not.

Acknowledgements – was there no one who helped with the study, typing things up, or entering data, or facilitating access? Minor things not meriting authorship.

Authors Reply: All work related to this manuscript done by research team themselves

---

## [Editor Report · Decision Letter 2]

9 Nov 2020

Hepatitis B vaccination status and knowledge, attitude, and practice regarding Hepatitis B among preclinical medical students of a medical college in Nepal

PONE-D-20-25582R2

Dear Dr. Shrestha,

We’re pleased to inform you that your manuscript has been judged scientifically suitable for publication and will be formally accepted for publication once it meets all outstanding technical requirements.

Kind regards,

Isabelle Chemin, PhD

Academic Editor

PLOS ONE
---

## [Editor Report · Acceptance letter]

12 Nov 2020

PONE-D-20-25582R2 

Hepatitis B vaccination status and knowledge, attitude, and practice regarding Hepatitis B among preclinical medical students of a medical college in Nepal 

Dear Dr. Shrestha:

I'm pleased to inform you that your manuscript has been deemed suitable for publication in PLOS ONE. Congratulations! Your manuscript is now with our production department. 

Kind regards, 

on behalf of

Mrs Isabelle Chemin 

Academic Editor

PLOS ONE